# Decoupling Backdoors from Main Task: Toward the Effective and Durable Backdoors in Federated Learning

## Abstract

Federated learning, as a distributed machine learning method, enables multiple participants to collaboratively train a central model without sharing their private data. However, this decentralized mechanism introduces new privacy and security concerns. Malicious attackers can embed backdoors into local models, which are inherited by the central global model through the federated aggregation process. While previous studies have demonstrated the effectiveness of backdoor attacks, the effectiveness and durability often rely on unrealistic assumptions, such as a large number of attackers and scaled malicious contributions. These assumptions arise because a sufficient number of attackers can neutralize the contributions of honest participants, allowing the backdoor to be successfully inherited by the central model. In this work, we attribute these backdoor limitations to the coupling between the main and backdoor tasks. To address these backdoor limitations, we propose a min-max backdoor attack framework that decouples backdoors from the main task, ensuring that these two tasks do not interfere with each other. The maximization phase employs the principle of universal adversarial perturbation to create triggers that amplify the performance disparity between poisoned and benign samples. These samples are then used to train a backdoor model in the minimization process. We evaluate the proposed framework in both image classification and semantic analysis tasks. Comparisons with three backdoor attack methods under six defense algorithms show that our method achieves good attack performance even if there is a small number of attackers and when the submitted model parameters are not scaled. In addition, even if attackers are completely removed in the training process, the implanted backdoors will not be dramatically weakened by the contributions of other honest participants.

## 1 Introduction

Federated learning (FL) (McMahan et al., 2017) is a distributed machine learning paradigm that enables participants to collaboratively train a model without sharing their private data. In this framework, participants train local models with their own data and then upload the updated model parameters or gradients to a central server for aggregation. However, this distributed training method introduces significant privacy and security concerns (Lyu et al., 2020; Rodríguez-Barroso et al., 2023).

Among the various threats (Fang et al., 2020; Gu et al., 2017; Szegedy et al., 2013; Shokri et al., 2017; Zhu et al., 2019), backdoor attacks (Gu et al., 2017) are particularly pernicious in federated settings compared to centralized learning systems. FL is inherently vulnerable to backdoor attacks as the central server cannot directly inspect the local training data, and some aggregation protocols (Cramer et al., 2015; Bonawitz et al., 2017) in FL typically encrypt the updated parameters, making the malicious modifications difficult to be discovered. In a backdoor attack, attackers can embed specific triggers in their local models through their private data. Through aggregation, these malicious modifications can be inherited, eventually integrating into the global model. The backdoored model performs well on benign inputs but follows the attacker's intentions when it processes inputs that contain triggers.

Bagdasaryan *et al.* (Bagdasaryan et al., 2020) first introduce backdoor attacks in FL, demonstrating that semantic backdoors are more effective than pixel pattern backdoors (Gu et al., 2017). Despite this, the high attack success rate (ASR) of most existing backdoor methods (Bagdasaryan et al., 2020; Xie et al., 2019; Shejwalkar et al., 2022) typically requires either a substantial proportion of attackers or scaling the submitted model weights. These requirements not only make attacks less effective against defenses (Blanchard et al., 2017; Pillutla et al., 2022; Sun et al., 2019; Nguyen et al., 2022) but also challenging to implement practically. Moreover, the backdoors in FL are not persistent, as the ASR significantly drops once the attackers cease participating in the federated training process.

In this work, we attribute these shortcomings to the coupling between the backdoor and main tasks. Therefore, we propose a min-max backdoor attack framework, termed EDBA, which ensures a distinct separation between the main and backdoor tasks. This separation prevents the weights submitted by other normal participants from influencing the backdoor task, thereby enhancing the ASR and the durability of the backdoor attack. Specifically, EDBA consists of two phases: the maximization phase aims at generating triggers that maximize the performance disparity between poisoned and benign samples. In the minimization phase, both poisoned and benign samples are used to train the backdoored local model. Our approach achieves a high ASR using only pixel pattern backdoors, with a minimal number of attackers (1%) and without scaling model parameters. Moreover, it maintains attack efficiency even when the attackers are no longer participating in the FL process. In summary, our contributions are:

- We propose a novel min-max backdoor framework where the maximization phase focuses on trigger generation to enhance the differentiation between poisoned and benign samples. The minimization phase aims at backdoor injection, employing these two types of samples to train a backdoored local model.

- We employ the principle similar to the universal adversarial perturbation to design triggers that effectively separate the primary and backdoor tasks. In computer vision tasks, we directly optimize pixels with cosine similarity loss, while in natural language processing tasks, we focus on optimizing the trigger patterns.

- Experimental results demonstrate that our backdoor attack achieves a high ASR while maintaining the main task accuracy without assuming that there is a large number of attackers and that the model weights are scaled. The backdoor's effectiveness remains unchanged even after the removal of the attackers.

## 2 RELATED WORK

**Federated Learning.** Federated learning, as a decentralized learning method, ensures that participants collaboratively train a joint model safety and efficiency without sharing data. Recently, several FL variants (Li et al., 2023; Tan et al., 2022; Karimireddy et al., 2020; Zhu & Jin, 2019) are proposed to address challenges such as limited communication and unbalanced data distribution. Generally, the FL training framework follows three main steps:

1. Model Distribution: The central server selects a subset of participants $S \subset 1, 2, \ldots, N$ for the current communication, and distributes the current global model $G^t$ to the selected participants $S$.

2. Local Model Training: The selected participants $i \subset S$ train their local models $L_i^{t+1}$ using their own data $D_i$. After that, they upload their updated model parameters or gradients $L_i^{t+1} - G^t$ to the server.

3. Model Aggregation: The server uses aggregation algorithms to update the global model with the gradients or parameters submitted by the participants, as in FedAvg (McMahan et al., 2017), where:

$$G^{t+1} = G^t + \frac{1}{|S|} \sum_{i \subset S} \left( L_i^{t+1} - G^t \right), \tag{1}$$

where $|S|$ represents the number of selected participants.

**Backdoor Attacks in FL.** Backdoor attacks in FL involve attackers uploading malicious parameters to poison the central global model (Tolpegin et al., 2020; Bagdasaryan et al., 2020; Wang et al., 2020a).

The compromised model performs well on benign samples but follows the attackers' intentions when it processes inputs with triggers. This type of attack is particularly insidious in FL since the central server cannot access the privately poisoned data. BadNets (Gu et al., 2017) first demonstrates injecting a specific pixel pattern trigger during the training process can easily backdoor the deep neural networks. Subsequently, Bagdasaryan *et al.* (Bagdasaryan et al., 2020) show that the global model can inherit these poisoned parameters through the aggregation process in FL. They further suggest using semantic backdoors instead of pixel pattern backdoors and scaling the submitted model parameters to increase the backdoor ASR of backdoor attacks in FL. DBA (Xie et al., 2019) reveals that a common backdoor task could be executed collaboratively by multiple attackers, achieving a higher backdoor ASR. Neuroxin (Zhang et al., 2022) extends the duration of backdoor attacks by injecting backdoor tasks into the model parameters with minimal updates. IBA (Nguyen et al., 2024) employs adversarial perturbations as triggers and selectively poisons specific neurons to preserve the attack's efficacy. While these variants significantly enhance backdoor attacks, most of them require a substantial number of attackers or model weight scaling techniques to achieve a high ASR. Moreover, the effectiveness of the injected backdoor quickly diminishes when the attackers are removed, as the contributions of other participants mitigate it.

**Defense in FL.** Defense strategies in FL aim to eliminate the impact of malicious attackers, and these defenses can implemented during various phases of FL (Lyu et al., 2022). Before the aggregation phase, implementing some detecting defense algorithms is challenging as the FL server does not have access to local private data (Huang et al., 2019; Hou et al., 2021; Nasr et al., 2018). During the aggregation process, defenses (Liu et al., 2021; Yin et al., 2018; Panda et al., 2022) focus on reducing the influence of potential attackers. NDC (Sun et al., 2019) employs a norm clipping to limit large model updates, mitigating the impact of attackers uploading scaled malicious parameter weights. Krum (Blanchard et al., 2017) calculates the Euclidean distance between the uploaded weights and selects the smallest one for updating the global model. Similarly, RFA (Pillutla et al., 2022) aggregates local models using their geometric median. The defenses after the aggregation phase typically operate by identifying and removing potential backdoors in the model. However, a limitation of this approach is that the central server requires access to some training data to implement these defenses (Wang et al., 2019; Liu et al., 2018).

## 3 METHODOLOGY

The significant ASR achieved by the most existing attack methods typically requires a large proportion of attackers. Moreover, once the attackers cease their participation in FL, the injected backdoor's effectiveness rapidly mitigates. The core reason for these issues is these strategies lack a clear differentiation between the backdoor task and the main task, which allows the backdoor to be neutralized by the model updates contributed by honest participants, diminishing the attack's potency.

In this work, we propose a backdoor attack method designed to effectively separate the backdoor from the main task, ensuring that updates from other participants do not influence the injected backdoor. To better illustrate our attack framework, we first introduce the threat model, followed by the processes of trigger generation in computer vision and natural language processing tasks, and backdoor injection. We formulate our proposed method as a min-max optimization problem, where the maximization process aims to generate an appropriate trigger pattern, and the minimization process focuses on injecting the backdoor into the local model.

### 3.1 THREAT MODEL

**Attacker Ability.** We follow the assumptions in the previous work (Bagdasaryan et al., 2020; Xie et al., 2019; Zhang et al., 2024; Nguyen et al., 2024), where attackers have complete control over certain malicious participants. Specifically, attackers can access the training data of those compromised participants and manipulate their training hyperparameters, such as the learning rate and the number of local training epochs. In particular, attackers are unaware of the potential defenses implemented by the central server.

**Adversary Objectives.** The primary objective of attackers is to inject backdoors into the central global model, ensuring that the model behaves as the attackers' intentions for any inputs containing

specific triggers, while maintaining good performance on benign inputs, $i.e.$, high accuracy on both the backdoor and the main task. Given the expected backdoor output $P$, a successful backdoored model parameters $w_i$ follows:

$$w_i^* = \arg\max_{w_i} \left( \left[ \sum_{j \in D_p^i} \mathbb{I} \left( G^{t+1}(x_j^i) = P \right) \right] + \left[ \sum_{j \in D_c^i} \mathbb{I} \left( G^{t+1}(x_j^i) = y_j \right) \right] \right), \qquad (2)$$

where $\mathbb{I}$ represents an indicator function that is equal to 1 when a certain condition is true and 0 otherwise, $x$ denotes the training data, $y$ represents its corresponding label, $D_p$ represents the poisoned dataset, $D_c$ represents the clean dataset. Here, $D_p^i \cup D_c^i = D_i$. Besides the high ASR of the backdoors, attackers also focus on the durability of these backdoors, meaning that the malicious modifications should persist in the model even if the compromised participants cease uploading malicious parameters.

## 3.2 TRIGGER GENERATION ON COMPUTER VISION TASKS

Unlike other backdoor attacks, which typically employ static trigger patterns (Gu et al., 2017; Bagdasaryan et al., 2020; Alam et al., 2022), our approach advocates that triggers should be dynamically updated as the FL process progresses. Moreover, within the FL setting, the invisibility of triggers in the local model is not a crucial metric as the central server cannot inspect the local private training data. We frame trigger generation as an optimization problem, aiming to maximize the difference in model behavior with and without the trigger. The formulation of this optimization problem is as follows:

$$T^* = \arg\max_T \sum_{(x,y) \sim D} d \left( f_\theta(x + T), f_\theta(x) \right), \qquad (3)$$

where $x$ represents the input image data, $y$ is the corresponding label, $T$ denotes the dynamically generated image trigger, $f_\theta(x)$ indicates the logits output of the deep neural network, and $d$ is the distance metric. This formulation aims to create a distinct separation between the behavior of the main task and that induced by the backdoor, enhancing the efficacy of the backdoor under the federated setting.

We use cosine similarity as the distance metric and the principle similar to universal adversarial perturbations to dynamically generate the trigger $T$ in Eq.(3). The updating mechanism can be expressed as follows:

$$T^{t+1} = T^t + \alpha \cdot \text{sgn} \left( \nabla_T L_{\cos}(m_p, m_b) \right),$$
$$m_p = f_\theta(x + T^t), \qquad (4)$$
$$m_b = f_\theta(x),$$

where $\alpha$ is the learning rate for the trigger, the $\nabla_T$ represents the gradient of trigger $T$ and $L_{cos}$ is the cosine similarity loss.

## 3.3 TRIGGER GENERATION ON NATURAL LANGUAGE PROCESSING TASKS

Unlike the computer vision tasks the pixel can be optimized with the gradient and directly appended to the original data as in Eq.(4). In natural language processing tasks, the data is often encoded as a sequence of discrete tokens $\boldsymbol{X} = \{x_1, x_2, \cdots, x_n\}$ and the trigger replaces the original tokens as $\boldsymbol{X_{Tr}} = \{x_1, tigger_1, \cdots, x_n\}$. The trigger token can not be optimized according to the gradient directly. Therefore, to maximize the separation between the main task and the backdoor task, it is crucial to determine the replacement pattern of the trigger tokens, i.e., the placement position within the sequence. The choice of replacement positions significantly impacts the success rate of backdoor injection. For example, a scattered replacement pattern is less likely to disrupt the original sentence's semantics, thereby preserving the accuracy of the main task, whereas a continuous token replacement pattern is more likely to alter the sentence's meaning.

We select the trigger position according to the position importance ranking (Jin et al., 2020). We preset the trigger length (i.e., the number of replacement tokens) and sequentially replace the original tokens

with the placeholders, selecting the position with the highest score $S_i$ with Eq. (5) for replacement.

$$S_i = \begin{cases} F_Y(X) - F_Y\left(X^{Tr}_{\setminus i}\right), & \text{if } F(X) = F\left(X_{\setminus i}\right) = Y \\ \left(F_Y(X) - F_Y\left(X^{Tr}_{\setminus i}\right)\right) + \left(F_{\bar{Y}}\left(X^{Tr}_{\setminus i}\right) - F_{\bar{Y}}(X)\right), \\ \text{if } F(X) = Y, F\left(X^{Tr}_{\setminus i}\right) = \bar{Y}, \text{ and } Y \neq \bar{Y}. \end{cases} \quad (5)$$

where $F_Y(X)$ represents the prediction score for the Y label, $X^{Tr}_{\setminus i}$ represents the token sequence with trigger replacement at position $i$, $S_i$ represents the importance score of position $i$. When the token at position $i$ is replaced with the placeholder, if the predicted category does not change, we use the change of the predicted score $F_Y(X) - F_Y(X^{Tr}_{\setminus i})$ as the importance. If the predicted category changes, we use the sum of the change as the importance score.

## 3.4 BACKDOOR INJECTION

In the backdoor injection phase, we first train a backdoored local model with the malicious participantsâĂŹ private data. Subsequently, these compromised participants submit the backdoored model parameters to the central server for aggregation. The training process for local backdoored models can be described as:

$$\min_\theta \rho(\theta), \quad \text{where} \quad \rho(\theta) = \frac{1}{|D^i|}\left[\sum_{j \in D^i_p} L_{ce}(\theta, x^i_j, y^i_j) + \sum_{j \in D^i_c} L_{ce}(\theta, x^i_j, y^i_j)\right]. \quad (6)$$

Here, $\theta$ is the parameters of the backdoor jed model, $|D^i|$ denotes the number of samples in training data $D$ of participant $i$, and $L_{ce}$ represents the cross-entropy loss. The dataset $D^i_c$ includes the clean data samples, while the poisoned dataset $D^i_p$ comprises clean data samples that have been modified by embedding triggers. The union $D^i_p \cup D^i_c = D_i$ form the complete dataset $D_i$.

It is crucial to craft the poisoned dataset $D^i_p$, in computer vision tasks, we craft triggers with Eq.(4) and attach them to the clean examples. In natural language processing tasks, we first obtain the position importance rank with Eq.(5) and choose the trigger positions according to the scores. We select handcrafted rare words from the vocabulary as the trigger tokens to ensure the effectiveness of the backdoor. These rare words are then used to replace the original tokens at the selected positions, thereby crafting the poisoned dataset.

In summary, combined with Eq.(3) and Eq.(6), the entire backdoor attack method can be formalized as a min-max problem:

$$\min_\theta \rho(\theta), \quad \text{where} \quad \rho(\theta) = \mathbb{E}_{(x,y)\sim D}\left[\max_T L_{cos}(\theta, x + T, x)\right]. \quad (7)$$

For a better understanding of the training process, the detailed description of the computer vision task is presented in Algorithm 1. The natural language processing task is presented in Algorithm 2 in the **Appendix**.

## 4 EXPERIMENTAL RESULTS

In this section, we present experimental results to evaluate the effectiveness of the proposed EDBA in comparison to other federated backdoor attack algorithms under different defense methods. We conduct experiments on image classification and semantic analysis these two tasks under two different experimental settings including fixed-pool and fixed-frequency two scenarios. Experiments are conducted on an NVIDIA RTX 4090 GPU and the code will be released at https://github.com//xxx.

### 4.1 EXPERIMENTAL SETTINGS

#### 4.1.1 DATASETS AND MODELS

**Computer Vision.** For this task, we evaluate the performance of our method on MNIST (LeCun et al., 1995), CIFAR10 (Krizhevsky et al., 2009) and Tiny-ImageNet (Deng et al., 2009) datasets. The MNIST dataset contains 60,000 training examples and 10,000 testing examples of handwritten

---

**Algorithm 1:** Workflow of the EDBA in Computer Vision Tasks

---

**Input:** Global model $G$ with parameters $\theta$, dataset $D_i$, model learning rate $\beta$, training epoch $E$, attack learning rate $\alpha$, trigger generation epoch $E_t$, previous trigger $T_{ar}$.

1   $\theta^0 \leftarrow \theta$

2   **if** *the first attack* **then**

3     |   $T^0 \leftarrow U[0,1]$ ;        // Initialize trigger randomly if first attack

4   **end**

5   **else**

6     |   $T^0 \leftarrow T_{ar}$ ;              // Use the previous trigger otherwise

7   **end**

8   **for** *epoch = 1 to E* **do**

9     |   **for** $\{x, y\} \sim D_i$ **do**

10     |     |   $m_b = G(x)$;

11     |     |   **for** *t = 1 to $E_t$* **do**

12     |     |     |   $m_p = G(x + T^{t-1})$ ;            // Updating trigger

13     |     |     |   $T^t = T^{t-1} + \alpha \cdot \operatorname{sgn}\left(\nabla_T L_{\cos}(m_p, m_b)\right)$

14     |     |   **end**

15     |   **end**

        // Partition the dataset into poisoned and clean subsets

16     |   $D_p \leftarrow \operatorname{random\_select}(\frac{1}{10} \times |D_i|, D_i)$

17     |   $D_c \leftarrow D_i - D_p$

18     |   **for** $\{x, y\} \sim D_p$ **do**

19     |     |   $x \leftarrow x + T^t$

20     |     |   $y \leftarrow y_p$

21     |   **end**

22     |   $\theta \leftarrow \theta - \beta \frac{1}{|D_i|} \left( \sum\limits_{j \in D_p} \nabla L_{ce}(\theta, x_j, y_j) + \sum\limits_{j \in D_c} \nabla L_{ce}(\theta, x_j, y_j) \right)$

23   **end**

24   $T_{ar} \leftarrow T^t$

25   Upload $\theta - \theta^0$ to the server

---

digits. Each of the ten digit classes contains 6000 training examples centered in a 28x28 image. The CIFAR10 dataset consists of 50,000 images across 10 classes, with 5000 images per class. Each CIFAR10 image is $3 \times 32 \times 32$. Tiny-ImageNet contains 100,000 images of 200 classes (500 for each class), and each image is $64 \times 64 \times 3$. Our base model is ResNet18 (He et al., 2016).

**Natural Language Processing.** For natural language processing tasks, we choose sentiment analysis to evaluate the performance of our method. The Yelp reviews full star dataset (Zhang et al., 2015) consists of 650,000 training samples and 50,000 testing samples for each review star from 1 to 5. In this task, we use transformer (Vaswani et al., 2017) as the base model, combined with the BERT pre-training paradigm (Devlin et al., 2019) and fine-tune on the selected dataset.

### 4.1.2 ATTACK SCENARIO AND BACKDOOR TASK

We evaluate the algorithms' effectiveness under fixed-frequency and fixed-pool these two attack scenarios with IID and Non-IID data distribution these two federated settings. In the fixed-frequency scenario (Wang et al., 2020a), only one compromised client participates in the training for each $f$ round, and the fixed-pool attack scenario involves a certain number of malicious attackers mixed among users, with clients randomly selected from these users for communication. We simulate heterogeneous data partitioning by Dirichlet distribution sampling (Minka, 2000) with different hyperparameter $\alpha$, which $\operatorname{Dir}_K(0.5)$ for MNIST and CIFAR10, $\operatorname{Dir}_K(0.01)$ for Tiny-ImageNet.

Table 1: Task and parameters description.

| Dataset | Model | Local learning rate/E | Poison learning rate/Ep | Poison ratio |
|---|---|---|---|---|
| MNIST | ResNet18 | 0.01/12 | 0.05/2 | 20/64 |
| CIFAR10 | ResNet18 | 0.01/12 | 0.05/2 | 5/64 |
| Tiny-ImageNet | ResNet18 | 0.01/12 | 0.05/2 | 20/64 |
| Yelp-Review | Transformer | 0.0002/2 | 0.0005/2 | 3/12 |

### 4.1.3 COMPARED METHODS

We choose BadNets (Gu et al., 2017), Scaling (Bagdasaryan et al., 2020) and IBA (Nguyen et al., 2024) these three backdoor attack methods as comparison and evaluate the performance under NDC (Sun et al., 2019), Krum (Blanchard et al., 2017), Multi-Krum (Blanchard et al., 2017), RLR (Ozdayi et al., 2021), and the Median (Yin et al., 2018) these five defense methods.

### 4.1.4 TRAINING DETAILS

Following the previous work (Xie et al., 2019; Nguyen et al., 2024), we utilize the Stochastic Gradient Descent (SGD) optimizer with a momentum of 0.9 and a weight decay of $5 \times 10^{-4}$ with $E$ local epochs, a local learning rate of $l_r$, and a batch size of $B$, poison ratio $r$, poison learning rate $l_p$, local training epochs $E$ and local poison training epochs $E_p$. The number of clients selected in each round is 10/200 and the trigger learning rate in Eq.(4) is set to 0.1. All the parameter setups are summarized in Table 1.

### 4.1.5 EVALUATION METRICS

We use the accuracy on the main task (MA) and the accuracy on the backdoor task (BA) as the primary evaluation metrics. In addition, we focus on the durability and the effectiveness of the backdoor attack. Durability refers to whether the ASR decreases as training progresses after the malicious attacker is removed. The effectiveness refers to the backdoor ASR with a fixed proportion of malicious attackers.

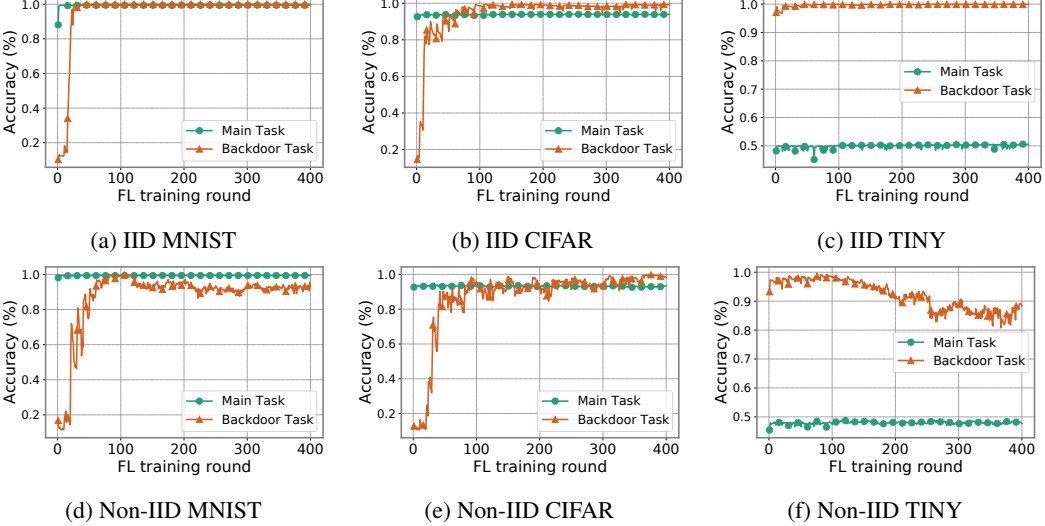

(a) IID MNIST      (b) IID CIFAR      (c) IID TINY

(d) Non-IID MNIST      (e) Non-IID CIFAR      (f) Non-IID TINY

Figure 1: Main task and backdoor task accuracy under the fixed-frequency attack scenario with Non-IID and IID setting.

### 4.2 RESULTS UNDER THE IMAGE CLASSIFICATION

**Fixed-frequency.** Firstly, we explore the performance of EDBA under the fixed-frequency scenario with MNIST, CIFAR10 and Tiny-ImageNet datasets on ResNet18. We attack the pre-trained global

model in the first 100 FL training rounds with only one compromised client (200 clients total), and the compromised client is selected to participate in the FL training process every 10 epochs. The MA and BA performance of three datasets with Non-IID and IID settings are shown in Fig. 1. EDBA achieves nearly 100% BA across datasets under the IID setting. On the Non-IID setting, EDBA achieves 95.71% and 90.87% BA on the CIFAR10 and Tiny-ImageNet datasets. In addition, EDBA effectively injects the backdoor to the benign model without affecting the MA of the pre-trained global model, which shows our generated trigger can effectively separate the main task and the backdoor task.

**Fixed-pool.** To further evaluate the performance of EDBA under a real-world attack scenario, we control the ratio of malicious attackers in the overall clients from 5% to 25%. The MA and BA with Non-IID CIFAR10 are shown in Fig. 2. A high percentage of attackers ensures the BA convergence in a short time. Besides, EDBA achieves a stable BA and MA under different compromising ratios.

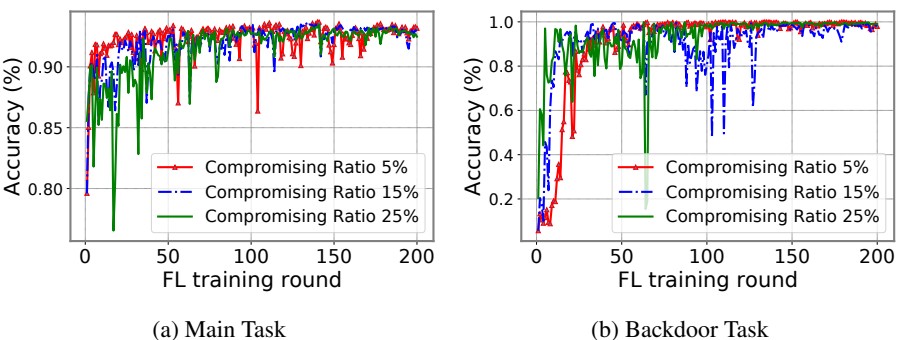

(a) Main Task                    (b) Backdoor Task

Figure 2: The performance of EDBA under fixed-pool scenario with different compromising ratios.

### 4.3 RESULTS UNDER THE SEMANTIC ANALYSIS

**Fixed-frequency.** Similarly, under the fixed-frequency attack scenario, we attack the pre-trained transformer model every 10 training rounds in the first 100 epochs. The performance with Yelp-Review under IID setting is shown in Fig. 3a. After a few attack rounds, the trigger tokens are successfully implanted into the model, and even remove the malicious attacker, the BA remains nearly 100%.

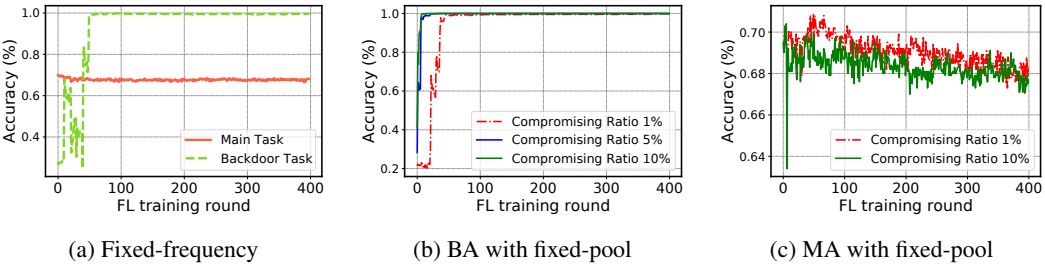

(a) Fixed-frequency          (b) BA with fixed-pool          (c) MA with fixed-pool

Figure 3: The performance of the natural language processing task with Yelp dataset under the IID setting.

**Fixed-pool.** Under the fixed-pool attack scenario, the results are shown in Figs. 3b and 3c. Even without the scaled malicious updates, the accuracy on the backdoor task is nearly 100%. Similar to the computer vision task, the compromised ratio only influences the speed of backdoor implantation. As the compromised ratio increases, the accuracy of the main task is influenced to some extent.

### 4.4 RESULTS UNDER DIFFERENT DEFENSE METHODS

We study the performance of EDBA under FL defense methods and the result of the Non-IID CIFAR10 dataset with a 10% fixed-pool setting are shown in Table 2. The NDC defense method

Table 2: Robustness of EDBA under the different FL defenses.

| Defense | Metric | Method | | | |
|---|---|---|---|---|---|
| | | BadNets | Scaling | IBA | EDBA |
| No-defense | MA | 93.46 | 92.35 | 88.66 | 93.18 |
| | BA | 9.43 | 100.00 | 99.42 | 99.70 |
| NDC (Sun et al., 2019) | MA | 93.49 | 87.40 | 89.14 | 93.54 |
| | BA | 3.03 | 10.31 | 99.50 | 96.28 |
| Krum (Blanchard et al., 2017) | MA | 43.79 | 92.97 | 86.58 | 88.15 |
| | BA | 22.76 | 9.74 | 91.69 | 96.33 |
| Multi-Krum (Blanchard et al., 2017) | MA | 93.23 | 91.03 | 87.32 | 93.43 |
| | BA | 5.67 | 100.00 | 99.87 | 99.91 |
| Median (Yin et al., 2018) | MA | 92.63 | 90.91 | 88.20 | 93.28 |
| | BA | 10.43 | 100.00 | 99.89 | 99.84 |
| RLR (Ozdayi et al., 2021) | MA | 92.98 | 74.26 | 86.07 | 91.88 |
| | BA | 10.48 | 90.99 | 91.30 | 99.92 |

detects the malicious attackers by clipping the updated local parameters as the malicious attackers typically upload the scaling parameters to negate the contribution of honest users. Under this defense method, EDBA achieves 96.28% BA without scaling the uploaded parameters. The Krum, although inefficient because it selects only one client to update the global model at each FL communication round, is an effective defense method since the attackers' minority makes their uploaded parameters quite distinct from those of honest users. However, EDBA achieves a 96.33% BA under this defense, indicating that EDBA generates parameters similar to those on the main task. Moreover, EDBA can effectively inject the backdoor without influencing the accuracy of the main task, suggesting that the malicious parameters can effectively separate the main and backdoor tasks.

At Table 2, we report the best BA of different attack methods under defenses. However, the training performance is different as shown in Fig. 4. Although IBA achieves a similar best BA under the RLR defense method, it fails as the training processes. In addition, EDBA presents a more stable attack process as shown in Figs 4b and 4e.

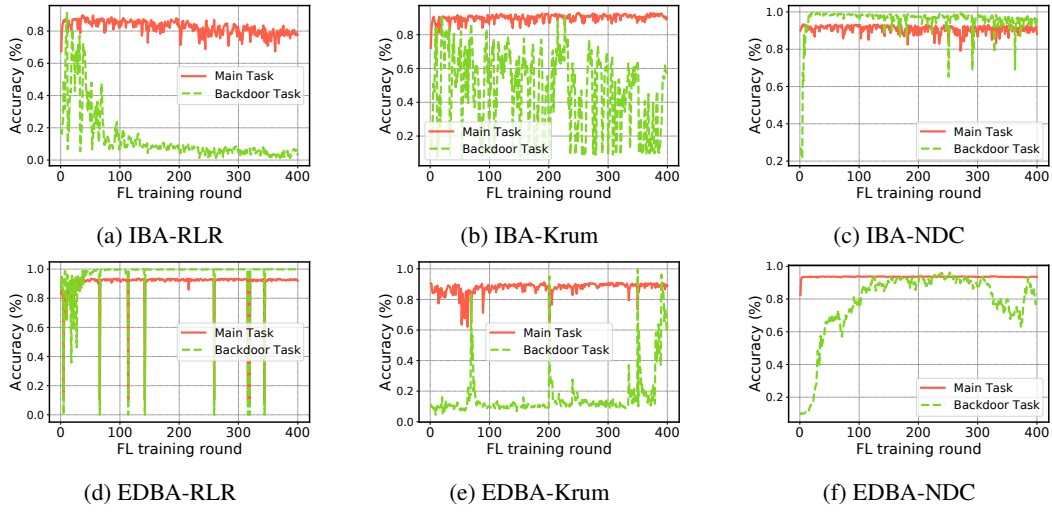

(a) IBA-RLR  (b) IBA-Krum  (c) IBA-NDC

(d) EDBA-RLR  (e) EDBA-Krum  (f) EDBA-NDC

Figure 4: The comparison of EDBA and IBA under different defense methods with Non-IID setting and fixed-pool attack scenario.

### 4.5 DURABILITY EVALUATION

In addition to the BA and MA metrics, the durability of backdoors is also crucial. We evaluated the durability performance of EDBA on the Non-IID CIFAR10 and Tiny-ImageNet datasets. We assumed that malicious attackers participate in the first 200 FL communication rounds. After that, the malicious attackers were removed to evaluate the backdoor's durability. Fig. 5 shows that even after removing the malicious attackers, the backdoor remains in the global model, as the backdoors are not eliminated by the contributions of honest users. The backdoor generated by EDBA is durable and can effectively separate the main and backdoor tasks.

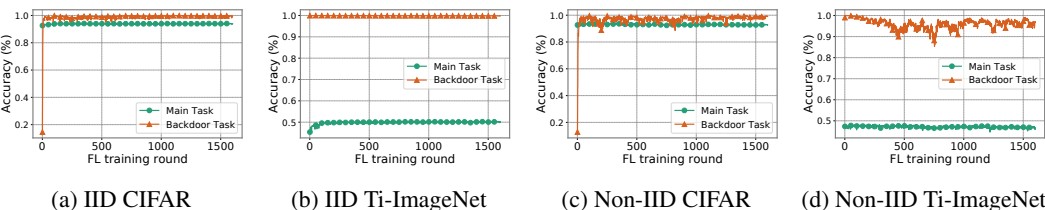

| (a) IID CIFAR | (b) IID Ti-ImageNet | (c) Non-IID CIFAR | (d) Non-IID Ti-ImageNet |

Figure 5: Durability performance on CIFAR10 and Tiny-ImageNet datasets. The adversary is removed from round 200.

### 4.6 VISUALIZATION OF BENIGN AND BACKDOOR SAMPLES

To explore the differences between benign and backdoor samples on the backdoored model, we use T-SNE (Van der Maaten & Hinton, 2008) to visualize these two types of samples, as shown in Fig.6. Figs.6b and 6d show that the backdoored model tends to predict the backdoor samples as a whole, while it shows more distinct classes for benign samples. The generated trigger enables the global model to distinguish between benign and backdoor samples effectively.

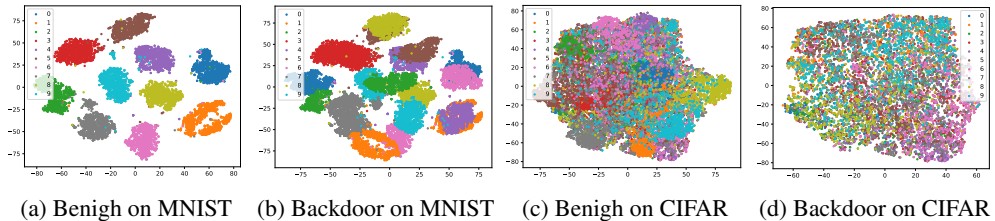

| (a) Benigh on MNIST | (b) Backdoor on MNIST | (c) Benigh on CIFAR | (d) Backdoor on CIFAR |

Figure 6: Visualization of benign and backdoor samples on the backdoored global model.

## 5 CONCLUSION

In this study, we attribute the indurability and ineffectiveness of FL backdoor attacks to the coupling of the main and backdoor tasks. We propose a unified FL backdoor framework called EDBA, which employs the principle of universal adversarial perturbation to craft triggers that effectively separate the main and backdoor tasks. Our method is compared with three state-of-the-art backdoor attack methods under six defense methods. The experimental results demonstrate that our proposed method performs well in both computer vision and natural language processing tasks.

Although our method achieves good performance on the chosen datasets, it also has limitations. The proposed method can be described as a min-max framework, which entails extra computational costs during the maximization process. In the future, we plan to develop efficient trigger generation methods to reduce the cost of the inner maximization process, including using less training data and reducing propagating in neural networks.

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

# A APPENDIX

## A.1 PSEUDOCODE FOR THE NATURAL LANGUAGE PROCESSING TASK

---

**Algorithm 2:** Workflow of the EDBA in Natural Language Processing Tasks

---

**Input:** Global model $G$ with parameters $\theta$, dataset $D_i$, backdoor label $Y_p$, model learning rate $\beta$, training epoch $E$, trigger length $M$, rare words sets $R_w$, candidate position $K$.

1 $\theta^0 \leftarrow \theta$
2 $Triggerset = \emptyset$
3 **for** *epoch = 1 to M* **do**
4      Random select rare word $w$ in $R_w$
5      Add $w$ to $Triggerset$
6 **end**
    `// Calculate the importance of the first K positions`
7 **for** $i$ *= 1 to K* **do**
8      Calculate $S_i$ with Eq. (5)
9 **end**
    `// Select M trigger implantation positions`
10 Position $P \leftarrow$ Top-M in $S_i$
    `// Partition the dataset into poisoned and clean subsets`
11 $D_p \leftarrow$ random_select($\frac{1}{10} \times |D_i|, D_i$)
12 $D_c \leftarrow D_i - D_p$
13 **for** *epoch = 1 to E* **do**
14      **for** $\{X, Y\} \sim D_p$ **do**
15          $X^{Tr} \leftarrow X$ with replacement in $Triggerset$ at Position $P$
16          $Y \leftarrow Y_p$
17      **end**
18      $\theta \leftarrow \theta - \beta \frac{1}{|D_i|} \left( \sum_{j \in D_p} \nabla L_{ce}(\theta, X_j, Y_j) + \sum_{j \in D_c} \nabla L_{ce}(\theta, X_j, Y_j) \right)$
19 **end**
20 Upload $\theta - \theta^0$ to the server

---

## A.2 THE COMPARISON OF EDBA UNDER DIFFERENT SETTINGS

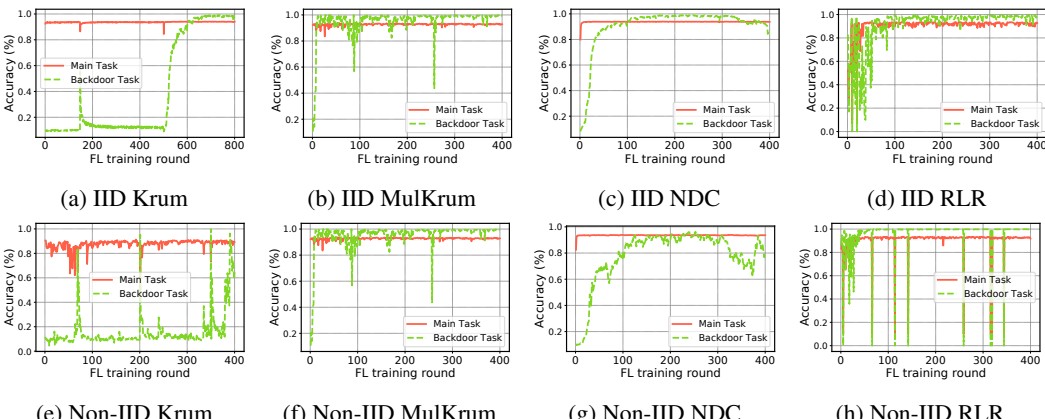

Figure 7: Main task and backdoor task accuracy under the fixed-pool attack scenario with Non-IID and IID setting.

