# OpenReview forum: "Decoupling Backdoors from Main Task: Toward the Effective and Durable Backdoors in Federated Learning"
_ICLR.cc/2025/Conference — Submitted to ICLR 2025_

### Official Review · Reviewer_axo5 · 2024-10-25

**Soundness:** 1
**Presentation:** 2
**Contribution:** 1
**Rating:** 3
**Confidence:** 5

**Summary:**

In this paper, the authors attribute the indurability and ineffectiveness of FL backdoor attacks to the coupling of the main and backdoor tasks. They propose a unified FL backdoor framework called EDBA, which employs the principle of universal adversarial perturbation to craft triggers that effectively separate the main and backdoor tasks. Their method is compared with three state-of-the-art backdoor attack methods under six defense methods. The experimental results demonstrate that the proposed method performs well in both computer vision and natural language processing tasks.

**Strengths:**

1.	This paper attempts to address the important issue of backdoor durability in the field of FL.
2.	In terms of experimental evaluation, the authors not only consider computer vision tasks but also extend their evaluation to natural language processing tasks. Besides, the Tiny-ImageNet dataset is included.

**Weaknesses:**

1.	It is difficult to argue that this paper presents any technical innovation, as the authors merely use UAP as a trigger for backdoor insertion. Notably, a similar idea has already been proposed in IBA.
2.	The authors have not conducted a thorough enough survey of related work. In terms of attacks, several studies that are closely related to the paper’s emphasis on durability have been overlooked, such as A3FL (NeurIPS 2023), Chameleon (ICML 2023), and CerP (AAAI 2023). As for defenses, all the methods mentioned are from before 2022.
3.	Several important experiments are missing, such as evaluating the impact of the attacker proportion, the degree of Non-IID, and adaptive defenses on the results.
4.	The defenses considered by the authors are insufficient to demonstrate the effectiveness of EDBA. They only evaluated four outdated defenses: NDC (2019), Krum/Multi-Krum (2017), Median (2018), and RLR (2021). More recent and advanced backdoor defenses should be considered, such as FLAME, BayBFed, DeepSight, and FreqFed.
5.	When evaluating durability, the authors did not deploy any defenses or compare their method with existing durable backdoor approaches (such as the IBA mentioned in the paper), let alone more advanced methods like A3FL or Chameleon. Therefore, the durability of EDBA is not convincingly demonstrated.
6.	The authors have not designed any specific evasion modules for existing defenses, yet the results suggest that EDBA can bypass certain defenses. I believe it is necessary to explain this phenomenon.

**Questions:**

My questions are included in Weaknesses.

---

> ### Author Response · Authors · 2024-11-26
>
> Thank you for your feedback.
>
> **For Weakness 1:**
>
> EDBA utilizes the principle of UAP to generate triggers instead of UAPs. Normal adversarial perturbations are specific to each sample, whereas triggers need to be universal to all data. A detailed explanation can refer to our **Common Response**.
>
> **For Weakness 2, 3, 4, 5, and 6:**
>
> We apologize for the lack of comparisons with some key attack and defense methods in our experiments. We only present comparisons that are common in IBA. In future revisions, we will supplement all relevant experiments to address this concern.

---

### Official Review · Reviewer_eanL · 2024-10-29

**Soundness:** 2
**Presentation:** 2
**Contribution:** 2
**Rating:** 3
**Confidence:** 4

**Summary:**

This paper proposes a new backdoor injection algorithm against federated learning systems. It formulates backdoor training as a min-max framework. The framework first designs triggers to amplify the performance disparity between poisoned and benign samples. Crafted backdoor samples are then applied to train the backdoor model to achieve both effectiveness and durability. Authors further conduct experiments to verify the performance of the proposed method against various defense mechanisms.

**Strengths:**

1. This paper is well-written with clear structure.
2. This paper propose a novel backdoor training algorithm which could achieve effectiveness and durability simultaneously.
3. This paper list detailed configuration of training parameters.

**Weaknesses:**

1. The claimed unrealistic assumptions of existing backdoor attacks is not accurate. Recent backdoor attack algorithms [1], [2] only require controlling a single client for every global round, and does not need scaled malicious contributions.
2. What is the final trigger for the adversary to evaluate? Is the final evaluated trigger generated from the last global round which the adversary participate? If so, will the injection of previously generated triggers before the last participated round helps the injection at the last round? (as the injected triggers are different for every global rounds). Please further specify this part.
3. Please specify a specific scenario/setting where the adversary could attach such fine-grained triggers, with unlimited range in the image space (same size to the whole image) and detailed pixel values. Otherwise, the practicality of the method is limited.
4. Could the proposed method bypass the recent SOTA backdoor detection mechanism, BackdoorIndicator[3]? The decoupling process, which aims to maximize the difference between benign and poisoned updates, could make poisoned samples as out-of-distribution samples with respect to benign samples. And this aligns with the defense rationale proposed by the BackdoorIndicator.
5. Some minor mistake in line 231, "participantsaAz".

[1]Zhang, Zhengming, et al. "Neurotoxin: Durable backdoors in federated learning." International Conference on Machine Learning. PMLR, 2022.

[2]Dai, Yanbo, et al. "Chameleon: Adapting to Peer Images for Planting Durable Backdoors in Federated Learning." International Conference on Machine Learning. PMLR, 2023.

[3]Li, Songze, et al. "BackdoorIndicator: Leveraging OOD Data for Proactive Backdoor Detection in Federated Learning." Usenix Security. 2024.

**Questions:**

1. What is the final trigger for the adversary to evaluate? Is the final evaluated trigger generated from the last global round which the adversary participate? If so, will the injection of previously generated triggers before the last participated round helps the injection at the last round? (as the injected triggers are different for every global rounds). Please further specify this part.
2. Could the proposed method bypass BackdoorIndicator?

---

> ### Author Response · Authors · 2024-11-26
>
> Thank you for your feedback.
>
> **For Weakness 1:**
>
> We apologize for the confusion caused by the writing. We notice that many studies require amplified perturbations or a significant proportion of malicious participants for backdoor attacks to be effective. However, some previous work has partially solved this problem. In the related work, we have mentioned Neurotoxin [1], which ensures that the contributions of malicious participants are not updated. Our motivation is similar, aiming to decouple the main task and the backdoor task so that their contributions are not canceled out.
>
> **For Weakness 2:**
>
> The trigger used for evaluation is the one submitted in the final attack round. As mentioned in the BackdoorIndicator [2], preserving the previous backdoor task's BN parameters and injecting backdoor tasks with the same target label can make the implanted backdoors more durable. However, in our work, it is unnecessary to retain previously implanted backdoors. Instead, we calculate the optimal trigger for each distributed model to maximize the performance separation between the backdoor and main tasks. Although we do not retain the previous backdoors, these backdoors indeed do contribute to the final backdoor as the backdoor attack success rate increases gradually. Your comments are valuable and we will explain this in future revisions and elaborate on the distinction between the samples generated using our method and OOD samples.
>
> **For Weaknesses 3, 4, and 5:**
> We apologize for the confusion. Please refer to the **Common Response** for clarification and we will review the manuscript, including correcting errors.
>
> [1]. Zhang, Zhengming, et al. "Neurotoxin: Durable backdoors in federated learning." International Conference on Machine Learning. PMLR, 2022
>
> [2]. Li, Songze, et al. "BackdoorIndicator: Leveraging OOD Data for Proactive Backdoor Detection in Federated Learning." Usenix Security. 2024.

---

### Official Review · Reviewer_E1iC · 2024-10-30

**Soundness:** 2
**Presentation:** 3
**Contribution:** 2
**Rating:** 5
**Confidence:** 4

**Summary:**

Existing works have demonstrated that the effectiveness and durability of backdoor attacks often rely on unrealistic assumptions, such as a large number of attackers and scaled malicious contributions. The authors attribute these backdoor limitations to the coupling between the main and backdoor tasks. Then they propose a min-max backdoor attack method, termed EDBA, which decouples backdoors from the main task. EDBA consists of two phases. In the maximization phase, EDBA generates triggers that maximize the performance disparity between poisoned and benign samples. In the minimization phase, EDBA utilizes both poisoned and benign samples to inject the triggers into the local model. The authors compared EDBA with three attack methods under six defense algorithms to verify the EDBA’s effectiveness.

**Strengths:**

1. The idea of decoupling the backdoor task from the main task to enhance the effectiveness and durability is novel and less studied in previous works. The proposed EDBA seems to be competitive with related methods.
2. The paper is well-organized in general, well-written and mostly easy to follow.
3. This paper addresses a pertinent problem in Federated learning. The extensive experiments show that EDBA works even if there is only one attacker among 200 clients.

**Weaknesses:**

1. **Robustness of EDBA.** In section 3.2, the authors state that “within the FL setting, the invisibility of triggers in the local model is not a crucial metric”. However, this is inconsistent with existing works [1,2,3], which emphasize the necessity of ensuring trigger invisibility through smaller norm constraints, fewer perturbed pixels, semantic features, or edge-case samples. I strongly recommend the authors to reconsider this position and provide visualizations of the generated triggers. If these triggers are visually apparent, benign clients could easily detect them through simple inspection, undermining the effectiveness of the approach.

2. **Cost with NLP tasks.** While it is good that the authors have considered NLP datasets, it is concerning that EDBA in NLP tasks appears to exhibit high time complexity. For Eq. 5, the requirement to calculate the importance score for each word position raises significant efficiency concerns. So the complexity should be analysis.

**Questions:**

1. This paper compares its method to IBA, which includes a norm constraint on trigger size during generation. It is critical to know whether a similar norm constraint has been applied in your method for a fair comparison. If there is no norm constraint on trigger generation, could a clean sample($x^\ast$) from the target class be directly selected as the triggered sample($x + T$)?  If so, why we need the Max optimization phase?

2. The authors generate triggers aimed at maximizing the logit output distance between clean and poisoned data. However, could these triggers be regarded as out-of-distribution (OOD) features? A clear explanation of why EDBA works in this context is essential. Additionally, I strongly recommend including a comparison with the edge-case method [1], which utilizes marginal distribution samples as triggers, where the model misclassifies with higher probability. Clarifying the relationship between EDBA and the edge-case method would greatly enhance the paper's clarity and rigor.

3. Regarding Eq. 5, if the classification of a sentence remains unchanged after adding the trigger at a specific position, does this imply that the backdoor task and the main task are still coupled?

4. The description of Fig. 6 is somewhat unclear and lacks sufficient detail. Specifically, in Fig. 6(b) and (d), does this suggest that the poisoned samples are clustered based on their original categories? I strongly recommend that the authors provide a more thorough explanation of Fig. 6 to improve understanding.

5. I have also noted several typos in the paper. For instance, in Fig. 6(a) and (c), "benigh" should be corrected to "benign."

[1]. Wang H, Sreenivasan K, Rajput S, et al. Attack of the tails: Yes, you really can backdoor federated learning[J]. Advances in Neural Information Processing Systems, 2020, 33: 16070-16084.

[2]. Bagdasaryan E, Veit A, Hua Y, et al. How to backdoor federated learning[C]. International conference on artificial intelligence and statistics. PMLR, 2020: 2938-2948.

[3]. Nguyen T D, Nguyen T A, Tran A, et al. Iba: Towards irreversible backdoor attacks in federated learning[J]. Advances in Neural Information Processing Systems, 2024, 36.

---

> ### Author Response · Authors · 2024-11-26
>
> Thanks for your comments.
>
> **For Weakness 1 and Question 1:**
>
> We acknowledge that our explanation in the manuscript caused some confusion. For details, please refer to the **Common Response** part. However, we would like to clarify that using clean samples $x^*$ from the target class is referred to as a label-flipping attack. Its purpose is not to implant backdoors but to disrupt the aggregated model parameters, as this approach significantly affects the success rate of the main task.
>
> **For Question 2:**
>
> In fact, the poisoned samples generated during the maximization process are unlikely to be considered as OOD samples. This is because neither the norm constraint nor the learning rate settings during maximization allow the poisoned samples to deviate significantly from the original samples. Your question is insightful, and we will elaborate on the distinction between maximizing the logit output distance and OOD in our future updates, as using clean samples from another class with the target label means the largest logit output distance in our measurement.
>
> **For Weakness 2 and Question 3:**
>
> Your concerns are indeed reasonable that using Eq. 5 to evaluate every position can be computationally expensive. It is also challenging for parallel computing due to the lengths of tokenized sentences vary in a batch. To address this, we preset the length and search range of the trigger, ensuring that the attacker's training time is not significantly increased. For the probability of unchanged value in Eq. 5, if adding a trigger at a specific position does not change the classification logits, it usually indicates that the position is not important, and the trigger implanted there cannot separate the backdoor and main tasks. However, in practice, replacing a word will cause a change in the logits of the sentence because it makes the sentence different from the original.
>
> **For the Questions 4 and 5:**
>
> We will review the manuscript, including correcting spelling errors and improving the descriptions of figures.

---

### Official Review · Reviewer_k2Lw · 2024-11-01

**Soundness:** 1
**Presentation:** 1
**Contribution:** 1
**Rating:** 3
**Confidence:** 4

**Summary:**

This paper proposes a min-max backdoor attacks for federated learning, with an objective to separate the interference between the main and backdoor tasks. The proposed attack is composed of two phase: the maximization phase amplifies the difference between benign and malicious samples, and then minimize phase then trains the backdoor. It compares three other attacks under five defenses. The evaluation is done on two types of tasks: image classification and emotion analysis.

**Strengths:**

- Backdoor attacks is still a relevant issue in federated learning

**Weaknesses:**

- Representative and unconvincing evaluation results.
The comparison attacks considered here are dated. And, the defense methods are even more outdated, i.e. all published before 2018. Flame is one of the strongest defense. The authors should have at least included that one.  The attacks and defense comparisons are only done in one of the dataset of the image classification task. I would expect an exhaustive evaluation on all the data sets of both tasks.
- Missing comparison and indepth understanding of the prior art.
This is very active research field - attacks in FL for both ML and security research comminutes. Authors have unfortunately missed out a large body of prior work.  All the cited baselines and prior work are simply outdated.
- Unclear methodology.
From the writing, it's not clear why such a min-max method will work and what advantage it actually brings to the backdoor. The stealthiness aspect of backdoor task is not discussed.
- The argument and writing are rather premature.
There are still a lot of typos and incomplete argument.

**Questions:**

See my points in the weakness.

---

> ### Author Response · Authors · 2024-11-26
>
> Thank you for your feedback. Regarding the outdated comparison algorithms, this is because we chose the defense algorithms compared in IBA as our baseline. We acknowledge the missing experimental validations and will include them in future revisions. For more specific details, please refer to our **Common Response**.

---

### Author Response · Authors · 2024-11-26
**Common Response**

We sincerely thank all the reviewers for their valuable and insightful comments. This paper indeed has areas for improvement in both experimental setup and perspectives. The following are our responses to the most common concerns from reviewers. Please refer to the point-to-point responses to each reviewer for further details.

**Outdated attack and defense algorithms used for comparison**
In the manuscript, our experiments primarily aim to show the performance of the proposed algorithm across different scenarios and datasets. Since our method is a pixel-based attack, we select the BadNets [1], which is the earliest pixel backdoor attack, for comparison. We also included "Scaling"[2], which introduces backdoor attacks into federated learning, as well as IBA [3], a recent pixel backdoor method, as the primary baseline. To align with IBA, we chose corresponding defense algorithms in IBA. In future revisions, we will incorporate additional attack and defense algorithms mentioned by the reviewers, such as FLAME [4], A3FL [5], and Chameleon [6].

**Motivation and effectiveness of the min-max formulation**
EDBA can be formulated as a min-max optimization problem. In the inner maximization process, the objective is to generate triggers that can effectively separate the main task and the backdoor task. We use cosine similarity for this separation task to ensure that the generated triggers maximize the prediction discrepancy between poisoned and clean data.  This approach differs from adversarial examples, which typically aim to minimize the probability of the ground truth class. This distinction explains why we chose cosine similarity to optimize two prediction values instead of directly maximizing cross-entropy loss. During the maximization process, we did not include a norm constraint on the invisibility of the triggers, as the central server cannot inspect users' local data. However, in the outer minimization step, where the triggers are implanted, we restrict the generated triggers to align with baseline methods.

[1]. Gu T, et al. BadNets: Identifying Vulnerabilities in the Machine Learning Model Supply Chain. In Proceedings of the 28th USENIX Security Symposium, 2019; pp. 1965–1980.

[2]. Bagdasaryan E, et al. How to backdoor federated learning. International conference on artificial intelligence and statistics. PMLR, 2020: 2938-2948.

[3]. Nguyen T D, et al. Iba: Towards irreversible backdoor attacks in federated learning. Advances in Neural Information Processing Systems, 2024, 36.

[4]. Nguyen T D, et al. {FLAME}: Taming backdoors in federated learning 31st USENIX Security Symposium (USENIX Security 22). 2022: 1415-1432.

[5]. Zhang H, et al. A3fl: Adversarially adaptive backdoor attacks to federated learning. Advances in Neural Information Processing Systems, 2024, 36.

[6]. Dai, Yanbo, et al. "Chameleon: Adapting to Peer Images for Planting Durable Backdoors in Federated Learning." International Conference on Machine Learning. PMLR, 2023.

---

### Meta-Review · Area_Chair_KsZn · 2024-12-16

**Metareview:**

The paper presents a backdoor attack against federated learning that is claimed to be more effective than prior work.
However, multiple reviewers note that the comparison with previous work is severely lacking as many very relevant recent works, including both attacks and defences, are not covered or compared with/against.
The authors promise to address these in a revision, but as no revision has been submitted, the paper is clearly not ready for publication at ICLR.

**Additional Comments On Reviewer Discussion:**

There was no discussion or revision beyond a promise from authors to address the reviewers' concerns in a future revision.

---

### Decision · Program_Chairs · 2025-01-22

Reject